# Localization Technique Using Mixture of Indigo Carmine and Lipiodol of Pulmonary Nodule via Bronchoscopic Navigation

**DOI:** 10.3390/medicina58091235

**Published:** 2022-09-06

**Authors:** Na-Hyeon Lee, Hyun-Sung Chung, Jeong-Su Cho, Yeong-Dae Kim, Jung-Seop Eom, Hyo-Yeong Ahn

**Affiliations:** 1Department of Trauma Surgery, Biomedical Research Institute, Pusan National University Hospital, Busan 49241, Korea; 2Division of Pulmonology, Center for Lung Cancer, National Cancer Center, Goyang 10408, Korea; 3Department of Thoracic and Cardiovascular Surgery, Biomedical Research Institute, Pusan National University Hospital, Busan 49241, Korea; 4Department of Internal Medicine, Biomedical Research Institute, Pusan National University Hospital, Busan 49241, Korea

**Keywords:** localization, pulmonary nodule, bronchoscopy, indigo carmine, lipiodol, video-assisted thoracoscopic surgery

## Abstract

*Background and Objectives*: As the number of minimally invasive surgeries, including video-assisted thoracoscopic surgery, increases, small, deeply located lung nodules are difficult to visualize or palpate; therefore, localization is important. We studied the use of a mixture of indigo—carmine and lipiodol, coupled with a transbronchial approach—to achieve accurate localization and minimize patient discomfort and complications. *Materials and Methods*: A total of 60 patients were enrolled from May 2019 to April 2022, and surgery was performed after the bronchoscopy procedure. Wedge resection or segmentectomy was performed, depending on the location and size of the lesion. *Results*: In 58/60 (96.7%) patients, the localization of the nodules was successful after localization, and 2/60 required c-arm assistance. None of the patients complained of discomfort during the procedure; in all cases, margins were found to be free from carcinoma, as determined by the final pathology results. *Conclusions*: We recommend this localization technique using mixture of indigo carmine and lipiodol, in concert with the transbronchial approach, because the procedure time is short, patient’s discomfort is low, and success rate is high.

## 1. Introduction

As chest computed tomography (CT) screening for lung cancer is performed more frequently, surgery for small pulmonary nodules is common [1]. Video-assisted thoracoscopic surgery (VATS) is a commonly used, minimally invasive technique; however, small, deeply located lung nodules are difficult to visualize and palpate during surgery, and localization is important for surgical success [2]. Although the position of the nodule is predicted before surgery through CT, the lung collapses during the operation, relocating the mass from the original predicted position [3]. If the exact position of the nodule is not known, the operation cannot be performed with VATS, and the procedure must be converted to an open thoracotomy [3,4]. Therefore, accurate localization of the pulmonary nodule is important for a high success rate in VATS [5,6].

Various localization methods have been developed, and they are largely classified according to their guidance systems and localization materials [3,6,7]. The main imaging tools in image-guided localization are CT or bronchoscopy, whereas localization materials are diverse and include metallic materials (hook-wire or microcoil), dyes (methylene blue or indigo carmine), contrast media (lipiodol, barium, or iodine contrast), or radiotracers (technetium-99m or 99mTc) [3,6,7].

Good localization techniques have a high success rate, stability, feasibility, and patient comfort [6,8]. To achieve these goals in our center, we decided to utilize a localization technique using a mixture of indigo carmine and lipiodol (MIL) administered via the transbronchial approach; here, we report feasibility, safety, and accuracy of the technique.

## 2. Materials and Methods

### 2.1. Ethical Statement

The patients provided written informed consent for the publication of clinical details and images. IRB No. 2003-027-088, the date of IRB approval: 30 March 2020.

This single center, retrospective, observational study was designed to use MIL administered via the transbronchial approach preoperatively to facilitate nodule localization during minimally invasive surgery.

### 2.2. Study Population (Patients)

From May 2019 to April 2022, 60 patients, who needed preoperative localization before diagnostic resection of a pulmonary nodule using VATS, met the inclusion criteria and were enrolled. The inclusion criteria were as follows: (1) in case the nodule type was solid, the minimum distance from the nearest pleural surface was greater than 10 mm. Pure ground-glass nodules or part-solid nodules were included regardless of the minimum distance from the pleural surface. (2) The maximum length of the nodule was less than 30 mm. (3) The target nodule was located in the peripheral third. Solid nodules, located less than 10 mm from the pleura, patients without informed consent, or nodules greater than 30 mm were excluded.

If the inclusion criteria were satisfied, the patient was informed of the advantages and disadvantages of the procedure, and bronchoscopy was performed on the morning of the day of surgery.

### 2.3. Marking Procedure

All bronchoscopic markings were performed the day of and prior to surgery. Before bronchoscopy, 0.5 cc indigo carmine and 0.5 cc lipiodol were mixed using a three-way stopcock and two syringes to thoroughly mix the two marking agents. Bronchoscopy was performed using a 4.0-mm flexible bronchoscope (BF-P260F; Olympus, Tokyo, Japan) under conscious sedation using fentanyl and midazolam. After airway inspection, the bronchoscope was advanced to the target lesion using virtual bronchoscopic navigation (Bronchus Technologies, Inc., Mountain View, CA, USA). Thereafter, under X-ray fluoroscopic guidance, a 20-MHz radial probe EBUS (UM-S20-17S; Olympus, Tokyo, Japan) was inserted into the guide sheath (K-201; Olympus, Tokyo, Japan) and pushed through the working channel towards and as close to the lesion as possible (Figure 1). After the lesion was targeted, the radial probe was removed, and MIL was injected through the guide sheath. Immediately after injection of the mixture, 2 cc of air was pushed into the guide sheath to spread the marker around the target lesion. After confirmation that the marking was successful using X-ray fluoroscopy, bronchoscopy was terminated. Flumazenil was administered after the procedure to keep the patient awake until surgery.

### 2.4. Endpoints

The primary endpoint was success rate, and success was defined as follows: (1) successful visualization such that C-arm use was deemed unnecessary, (2) brief time from port placement to initial resection, and (3) margins free of carcinoma, as determined by histopathology. The secondary endpoint was safety. Safety was defined as complications after localization, including post-procedural pain.

### 2.5. Statistical Analysis

Data were analyzed using the Statistical Package for Social Sciences^®^ software (SPSS^®^), version 22.0 (IBM^®^; Chicago, IL, USA), and statistical significance was defined as *p* < 0.05. For univariate statistical analysis, the unpaired Student’s *t*-test for clinical findings and chi-square test for gender distribution were used.

## 3. Results

Between May 2019 and April 2022, 60 patients met the inclusion criteria and underwent pulmonary localization using the preoperative transbronchial approach. Based on CT measurements, the average nodule size was 10.5 ± 2.8 mm (range, 5.8–16.0 mm), and the average nodule depth from the nearest pleura was 12.4 ± 10.2 mm (range, 1.2–41.5 mm). Of the 60 cases, nodules in 28 (46.7%) were solid type, 21 (35%) were part-solid, and 11 (18.3%) were the pure ground-glass type. The most common location of the nodule was the left upper lobe (24/60) (Table 1). The average time for the bronchoscopy marking procedure was 16 min (range, 5.1–100 min). There were no cases of post-procedural complications related to pulmonary nodule localization and marking, such as pneumothorax, bleeding, or respiratory failure (Table 2). In addition, patients did not report procedure-related pain or other types of discomfort during the procedure in which they were in a state of conscious sedation or right after the procedure when they were awake.

Patients were then moved to the operating room, and wedge resection or segmentectomy was performed, according to the depth of the lesion; if necessary, additional surgery (segmentectomy or lobectomy) was performed, according to the frozen section results, as definitive treatment (Figure 2). In 46 cases, the initial operation was wedge resection, and 14 patients underwent a segmentectomy. Of the 46 cases with wedge resection, a lobectomy was additionally performed in 20, and an additional segmentectomy was performed in 8. In addition, 5 out of 14 cases that underwent a segmentectomy as the initial operation were confirmed to be malignant, according to the frozen section results, and lobectomy was performed on the spot (Table 3). In 2/60 cases, visualization failed. When the thorcoscope was inserted during the operation, the marking material that would have had the blue color of indigo carmine was not seen in the parietal pleura; therefore, the location of the nodule was checked using a mobile C-arm and then resected. Comparing the successful and failed cases, the long and short diameter of the nodule, distance from the visceral pleura and second carina, and transbronchial diameter of pre- and post-nodule were not significantly different (*p* = 0.09, 0.11, 0.69, 0.34, 0.40, and 0.53). Nodules in the failed cases (*n* = 2) were in the left upper lobe (LUL); however, there was no significant difference in the location of the nodules (*p* = 0.35) overall between the successful and unsuccessful cases. The average time from the first detection of the nodule to initial resection was 29.1 ± 17.2 min (range, 5–69 min). For cases that took more than 40 min, adhesiolysis was performed for a long time before the initial resection. Intraoperative frozen section results revealed that, out of a total of 60 nodules, 10 nodules were diagnosed as benign lesions, with 50 nodules diagnosed as malignant. Benign lesions include hamartoma, plasma cell granuloma, benign anthracosis, and chronic granulomatous inflammation. Among malignancies, 10 nodules were metastatic and 40 were primary. Metastasis consisted of liposarcoma, renal cell carcinoma, and hepatocellular carcinoma. In cases of primary lung cancer, the results were as follows: adenocarcinoma, squamous cell carcinoma, small cell carcinoma, and large cell neuroendocrine carcinoma. There were no cases in which the benign or malignant tumor diagnoses were changed, based on the frozen results or final pathologic examination results; in all cases, the resection margin was confirmed to be free from carcinoma (Table 4).

The average length of hospital stay was 4.4 days, and the average length of the chest tube placement period was 1.6 days. There were no cases of perioperative complications, such as prolonged air leakage, bleeding, infection, pneumonia, or respiratory failure, after surgery.

## 4. Discussion

With the advent of lung cancer screening, subsolid (pure ground-glass or part-solid), solitary pulmonary nodules (SPNs) have been better detected and may or may not be excised for diagnosis. Particularly, solid SPNs greater than 8 mm should be biopsied via transbronchial or transthoracic routes [9]. However, transbronchial procedures may have low sensitivity and lack therapeutic potential. In addition, in a surgical approach via the transthoracic route, subsolid nodules, small solid nodules, or nodules 2 cm from the visceral pleura are difficult to visualize and palpate. In addition, there is a method to identify the lung nodule using an intra-thoracic ultrasound probe during surgery, but proper examination may be difficult, due to air in the lung parenchyma; if detection fails, conversion to thoracotomy may be required [10]. To overcome these problems, several techniques for pulmonary localization have been developed.

There are two categories of materials for pulmonary localization: directly visible and invisible, radio-opaque materials. Localization is achieved via either the transbronchial or transthoracic routes. Therefore, there are four ways to achieve pulmonary localization.

1Transthoracic marking using visible material, such as a hook or spiral wire [11,12].2Transthoracic dye marking.
ARadio-opaque materials: lipiodol [13].BRadio-lucent materials: indigo carmine, methylene blue, and indocyanin green [14].3Transthoracic radiotracer such as technetium-99m or 99mTc.4Transbronchial marking using visible material such as microcoil [15].5Transbronchial dye marking.
ARadio-opaque materials: lipiodol.BRadio-lucent materials: indigo carmine, methylene blue, indocyanin green.

Considering the above, transthoracic techniques, including transthoracic marking using a directly visible material and transthoracic dye marking, could develop complications, such as pneumothorax, hemothorax, wire migration, and pleuritic chest pain [16,17,18]. Hasegawa et al. used MIL as a material for localization in the same way as in our study, but reported that pneumothorax, hemothorax, and alveolar hemorrhage occurred as complications, due to the transthoracic approach [17]. Previously, our center also performed CT-guided lipiodol marking with a transthoracic approach to localize the pulmonary nodule. The success rate was about 90%, but the probability of pneumothorax after the procedure was close to 50%. In addition, pleuritic pain caused by leakage of lipiodol was confirmed in about 30% of cases. In particular, patients with COPD or emphysema required an oxygen supply, due to decreased saturation. Therefore, in order to relieve the patient’s pain and anxiety, the procedure was often performed within 1 to 2 h before surgery. In addition, radiation exposure of medical staff and patients was a problem because the C-arm had to be used to check the marking material in the operating room. If a radiotracer is used as a localization material, the location is confirmed using a gamma probe in the operating field. In this case, in order to place the radiotracer in an appropriate position, it is necessary to discuss the injection angle with the radiologist. In addition, surgery should be scheduled after the completion of CT-guided injections and follow-up scintigram in the radiology room [19]. Transbronchial marking using a directly visible material with intraoperative fluorography results in exposure to radiation for the patient and surgeons, and wire migration has been reported.

Transbronchial dye marking, using various radio-opaque and -lucent materials, is another option. In particular, using lipiodol injection followed by fluorography also has the drawback of radiation exposure [16,20,21]. Radio-lucent dyes, such as methylene blue or indigo carmine, are water soluble and diffuse easily, so accuracy decreased as the interval between injection and exposure increases [22,23,24]. To overcome problems caused by iatrogenic injury, such as pneumothorax and hemothorax to the lung, the transbronchial route is preferred over the transthoracic approach [25,26,27,28,29]. To overcome the problem of relatively fast diffusion from use of water-soluble dyes, we decided to use a high-viscosity material that could bind with dye and, therefore, extend the persistence of the dye at the lesion site after bronchoscopy. Recent research on long lasting materials has been published in Japan; however, in Korea, there is no qualification on the use of lipiodol under bronchoscopy, nor in the usage of MIL.

After the institutional review board of our hospital approved the mixtures for the research, we performed the described localization technique using MIL via the transbronchial approach in clinical practice. As mentioned in the results, our success rate was 96.7% (58/60), and no complications, such as pneumothorax, hemothorax, or pain, occurred. In the two failed cases, both patients had nodules located in LUL, which required an acute angle to approach, and CT showed that the peripheral bronchus behind the lesion might narrow, so the MIL could not be advanced. Therefore, in cases with nodules in the LUL and a narrow peripheral bronchus in which MIL cannot be advanced until exposure in the visceral pleura, we recommend lipiodol injection around the main mass, followed by indigo carmine injection near the visceral pleura, in order to allow for visualization by fluoroscopy.

Because hydrophobic and hydrophilic liquids cannot be easily commingled together, the MIL was made using a three-way stopcock and two syringes, and it was mixed 50 times to commingle the liquids together [30,31,32]. As the interval duration increased, the mixtures became separated, followed by the problem of excessive diffusion of the dye. To overcome this problem, we recommend using lidocaine gel to create micelles with the lipiodol and indigo carmine. If predicting the case with adhesion, which is assumed to have increased interval duration, MIL and lidocaine gel might be used in patients after clinical approval of pharmaceuticals in Korea. Although not found in our study, there may be spontaneously recovered inflammation, due to MIL and systemic embolization by the migration of the lipiodol component into the pulmonary vein; however, this has not been reported yet [17]. To prevent this, before injecting MIL, aspiration should be performed to confirm that it is not blood before injecting [17].

The limitation of this study is that the number of cases was small, and it was retrospective. The next step is to evaluate the effectiveness of this technique in a greater number of patients, as part of a prospective, randomized trial. In addition, the exact distance between the target lesions and sites marked by the localization technique could not be confirmed in the pathological examination. However, since the bronchoscope marking procedure is performed under conscious sedation, the volume of the lung changes with inspiration and expiration, as well as the position or range of distribution of the dye, may change during the operation as the lung is collapsed. Therefore, rather than measuring the exact distance between the dye marking and target lesion, it would be more appropriate to measure the presence of the target lesion in the specimen at the time of the first resection.

## 5. Conclusions

Since the localization technique using MIL via the transbronchial approach has the advantages of rapid administration, persistence of the marker at the lesion site, and decreased radiation exposure, we recommend it as an excellent option for localizing nodules. Although nodules located in the LUL with a narrow peripheral bronchus were more challenging, lipiodol might be injected around the main mass, followed by indigo carmine injection near the visceral pleura, so nodules could be double-checked by direct visualization, as well as fluoroscopy.

## Figures and Tables

**Figure 1 medicina-58-01235-f001:**
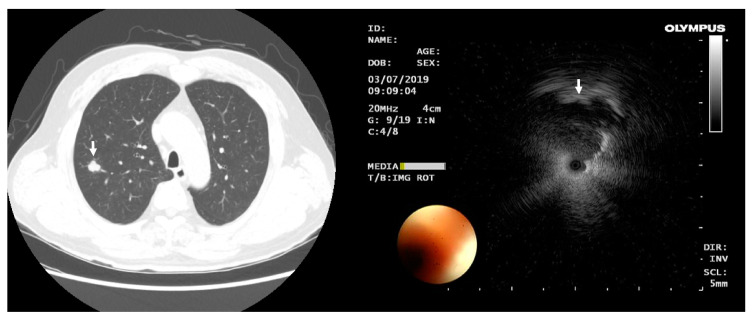
Preoperative CT scan and transbronchial marking procedure. The location and type of nodules (white arrow) was identified through CT before surgery; based on this, localization was performed using MIL via tranbronchial approach.

**Figure 2 medicina-58-01235-f002:**
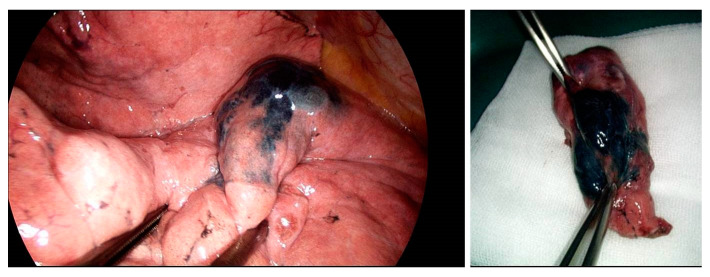
Video-assisted thoracoscopic image of nodule marked with MIL and resected lung. If localization is successful, the location can be easily identified through a video-assisted thoracoscopic image in the operating room.

**Table 1 medicina-58-01235-t001:** Clinical and radiological characteristics.

	N or Mean ± SD (Range)
**Total patients**	60
Age (y)	59.2 ± 11.0 (36–81)
Gender (male)	35
**Total nodules**	60
Nodule size (mm)	10.5 ± 2.8 (5.8–16.0)
Depth from pleura (mm)	12.4 ± 10.2 (1.2–41.5)
Location	
Right upper lobe	16
Right middle lobe	2
Right lower lobe	11
Left upper lobe	24
Left lower lobe	7
Nodule morphology	
Solid	28
Part-solid	21
Pure ground glass	11

**Table 2 medicina-58-01235-t002:** Procedural outcomes.

	N or Mean ± SD (Range)
**Success of localization**	58
**Average time of intervention (minutes)**	16.0 ± 20.2 (5.0–100)
**Complications**	
Pneumothorax	0
Bleeding	0
Respiratory failure	0
Air embolism	0
Premature termination	0

**Table 3 medicina-58-01235-t003:** Surgical approach and outcomes.

	N or Mean ± SD (Range)
**Initial operation type**	
VATS ^1^ wedge resection	46
VATS segmentectomy	14
**Final operation type**	
VATS wedge resection	18
VATS segmentectomy	17
VATS lobectomy	25
**Average length of hospital stay (days)**	4.4 ± 1.9 (2–9)
**Average length of the chest tube placement period (days)**	1.6 ± 0.8 (1–3)
**Postoperative morbidity**	
Air leakage over 4 days	0
Bleeding	0
Infection	0
Pneumonia	0
Respiratory failure	0

^1^ VATS (video-assisted thorcoscopic surgery).

**Table 4 medicina-58-01235-t004:** Pathologic characteristics.

	N
**Total nodules**	60
**Benign lesions**	10
Hamartoma	2
Plasma cell granuloma	1
Benign anthracosis	1
Chronic granulomatous inflammation	6
**Metastatic lesions**	10
Liposarcoma	1
Renal cell carcinoma	5
Hepatocellular carcinoma	4
**Primary lung cancer**	40
Adenocarcinoma	33
Squamous cell carcinoma	5
Small cell carcinoma	1
Large cell neuroendocrine carcinoma	1

## Data Availability

Not applicable.

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
