# Peer review of "Localization Technique Using Mixture of Indigo Carmine and Lipiodol of Pulmonary Nodule via Bronchoscopic Navigation"

_medicina, 2022, doi:10.3390/medicina58091235_

Round 1

Reviewer 1 Report

I read with great interest the proposed work for publication “. Localization technique using mixture of indigo-carmine and lipiodol of pulmonary nodule”.

The method described and applied on 60 patients seems very usefull and reproductible. The authors deserve congratulations for their scientific and medical work. The paper definitely has the necessary value for publication, also is well written. Below there are my suggestions as rewiever.

Title:

-        Bronchoscopy navigation or similar should be used in the title; so the fluoroscopic guidance. Otherwise the title may be incomplete and misleading.

Abstract:

-        Row 26 and 27: needs rephrasing. The surgery was performed anyway, so “the surgery was succesfull” is not correct, maybe the identification or the localisation of the nodule was succesfull.

Matherial and Method:

-        Inclusion criteria – please rephrase in order to understand if partial solide and GGO nodules are included

-        I suggest presenting the exclusion criteria – eg: less than 10mm from the pleura, lack of signed consent, nodules with 30mm or more etc.

-        Please specify the time between bronchoscopy and surgery, and the patient’s status (awake, sedated, intubated etc)

Results:

-        Rows 112-113: the average nodule depth – from the nearest pleura? Please specify. Even if there is specified in the discutions.

-        Again, the time between bronchoscopy and surgery is important, also the status of the patients – the patients experienced no pain if they were under sedation – please clarify that aspect, thank you.

-        Success of localisation – 58/60. Please clarify – the localisation failed. Please explain how do you identify the nodule? By identifying the color? after entering the chest or after resection? The methodology may not be clear for every reader, but this is easy to correct, and the results will be clearer after that.

-        “In addition, additional lobectomy was performed in 5 out of 14 cases in which segmentectomy  was performed as the initial operation (Table 3).” – please explain why. Rows 131-133.

Discutions:

-        There is also gamma-probe method for identifying lung nodules, also endothoracic ultrasound probe (not endobronchial) – so please reassess the literature and complete this part – is important because of its’ educational purpose.

-        Rows 185-201 – complications of other methods – well presented, please add references to support the affirmations.

-        Very good Discution chapter.

Conlusions – nothing to commedt.

References:

-        Please provide at least 30 references, there are a lot of publications in the literature.

Figures:

Figure 1: Please mark the nodule on the CT scan and on the EBUS image.

Author Response

1. Bronchoscopy navigation or similar should be used in the title; so the fluoroscopic guidance. Otherwise the title may be incomplete and misleading

→The title was revised to "Localization technique using mixture of indigo-carmine and lipiodol of pulmonary nodule via bronchoscopic navigation."

2. Row 26 and 27: needs rephrasing. The surgery was performed anyway, so “the surgery was succesfull” is not correct, maybe the identification or the localisation of the nodule was succesfull.

→ It was revised to "In 58/60 (96.7%) patients, the localization of the nodules was successful after localization".

3. Inclusion criteria – please rephrase in order to understand if partial solide and GGO nodules are included.

→ It was revised to "The inclusion criteria were as follows: (1) In case the nodule type is solid, the minimum distance from the nearest pleural surface is greater than 10 mm. Pure ground-glass nodules or part-solid nodules were included regardless of the minimum distance from the pleural surface"

4.  I suggest presenting the exclusion criteria – eg: less than 10mm from the pleura, lack of signed consent, nodules with 30mm or more etc.

→ "Solid nodules, located less than 10 mm from the pleura, patients without informed consent, or nodules greater than 30 mm were excluded." was added.

5. Please specify the time between bronchoscopy and surgery, and the patient’s status (awake, sedated, intubated etc)

→ "Flumazenil is administered after the procedure to keep the patient awake until surgery." was added.

6. Rows 112-113: the average nodule depth – from the nearest pleura? Please specify. Even if there is specified in the discutions.

→ It was revised to "the average nodule depth from the nearest pleura was 12.4 ± 10.2 mm".

7. Again, the time between bronchoscopy and surgery is important, also the status of the patients – the patients experienced no pain if they were under sedation – please clarify that aspect, thank you.

→ It was revised to "In addition, patients did not report procedure-related pain or other types of discomfort during the procedure in which they were in a state of under conscious sedation or right after the procedure when they were awake.".

8. Success of localisation – 58/60. Please clarify – the localisation failed. Please explain how do you identify the nodule? By identifying the color? after entering the chest or after resection? The methodology may not be clear for every reader, but this is easy to correct, and the results will be clearer after that.

→ It was revised to " In 2/60 cases, visualization failed. When the thorcoscope was inserted during the operation, the marking material that would have had the blue color of indigo-carmine was not seen in the parietal pleura; therefore, the location of the nodule was checked using a mobile C-arm and then resected.".

9. “In addition, additional lobectomy was performed in 5 out of 14 cases in which segmentectomy  was performed as the initial operation (Table 3).” – please explain why. Rows 131-133.

→ It was revised to "In addition, 5 out of 14 cases that underwent segmentectomy as the initial operation were confirmed to be malignant according to the frozen section results, and lobectomy was performed on the spot.".

10. There is also gamma-probe method for identifying lung nodules, also endothoracic ultrasound probe (not endobronchial) – so please reassess the literature and complete this part – is important because of its’ educational purpose.

→ "3. Transthoracic radiotracer such as technetium-99m or 99mTc" was added and the following contents were added.

1. Transthoracic marking using visible material such as a hook or spiral-wire [11,12]
2. Transthoracic dye marking
    A. Radio-opaque materials: lipiodol [13]
    B. Radio-lucent materials: indigo-carmine, methylene blue, indocyanin green [14]
3. Transthoracic radiotracer such as technetium-99m or 99mTc
4. Transbronchial marking using visible material such as microcoil [15]
5. Transbronchial dye marking

In addition, there is a method to identify the lung nodule using intra-thoracic ultrasound probe during surgery, but proper examination may be difficult due to air in the lung parenchyma, and if detection fails, conversion to thoracotomy may be required
. If a radiotracer is used as a localization material, the location is confirmed using a gamma probe in the operating field. In this case, in order to place the radiotracer in an appropriate position, it is necessary to discuss the injection angle with the radiologist. In addition, surgery should be scheduled after completion of CT-guided injections and follow-up scintigram in the radiology room.

11. Rows 185-201 – complications of other methods – well presented, please add references to support the affirmations.

→ Several references were added.

12. Please provide at least 30 references, there are a lot of publications in the literature.

→ Several references were added.

13. Please mark the nodule on the CT scan and on the EBUS image.

→ I changed the figure and added arrows to make the meaning clearer.

Reviewer 2 Report

First I would like to congratulate all the authors and the entire team on the completion of this study and manuscript. The authors discuss a new approach of using MIL via a transbronchial approach using advanced bronchoscope techniques like navigational branch and radial-EBUS. The results are quite promising with no complications reported. Authors do identify limitations being small sample size. my couple of concerns/comments are:

1. Hasegawa et. al (PMID: 30819492) did use MIL via thransthoracic approach before VATS resection and reported no complications due to MIL. It will be important to include this study and contrast to your approach in the discussion.

2. A lot of the text I read is without references, I agree there is not extensive literature, especially on these techniques and methods but should try to include some more references especially in the discussion part.

3. Authors could also comment on the generalizability of this technique. Also about using Lidocaine gel, has that used previously in such setting? any potential complications with it?

4. It it possible to include data from previous years from the same center when this new approach was not used and what were the success rates? I agree its not a statistically sound comparison but of there is a marked difference in success rates that should be highlighted.

Author Response

1. Hasegawa et. al (PMID: 30819492) did use MIL via thransthoracic approach before VATS resection and reported no complications due to MIL. It will be important to include this study and contrast to your approach in the discussion.

→ "Hasegawa et al. used MIL as a material for localization in the same way as in our study, but reported that pneumothorax, hemothorax, and alveolar hemorrhage occurred as complications due to the transthoracic approach" was added.

2. A lot of the text I read is without references, I agree there is not extensive literature, especially on these techniques and methods but should try to include some more references especially in the discussion part.

→ Several references were added.

3. Authors could also comment on the generalizability of this technique. Also about using Lidocaine gel, has that used previously in such setting? any potential complications with it?

→ "Although not found in our study, there may be spontaneously recovered inflammation due to MIL and systemic embolization by migration of the lipiodol component into the pulmonary vein, but this has not been reported yet [17]. To prevent this, before injecting MIL, aspiration should be performed to confirm that it is not blood before injecting." was added.

4. It it possible to include data from previous years from the same center when this new approach was not used and what were the success rates? I agree its not a statistically sound comparison but of there is a marked difference in success rates that should be highlighted.

→ "Previously, our center also performed CT-guided lipiodol marking with a transthoracic approach to localize the pulmonary nodule. The success rate was about 90%, but the probability of pneumothorax after the procedure was close to 50%. In addition, pleuritic pain caused by spileage of lipiodol was confirmed in about 30% of cases. In particular, patients with COPD or emphysema required oxygen supply due to decreased saturation. Therefore, in order to relieve the patient's pain and anxiety, the procedure was often performed within 1 to 2 hours before surgery. In addition, radiation exposure of medical staff and patients was a problem because the C-arm had to be used to check the marking material in the operating room." was added.

Round 2

Reviewer 2 Report

Thanks for adding the changes requested. I have no further concerns regarding the submission.